# Chemotherapeutic Strategies with Valnemulin, Tilmicosin, and Tulathromycin to Control *Mycoplasma hyopneumoniae* Infection in Pigs

**DOI:** 10.3390/antibiotics11070893

**Published:** 2022-07-04

**Authors:** Giovani Marco Stingelin, Marina Lopes Mechler-Dreibi, Gabriel Yuri Storino, Karina Sonalio, Henrique Meiroz de Souza Almeida, Fernando Antônio Moreira Petri, Luís Guilherme de Oliveira

**Affiliations:** School of Agricultural and Veterinarian Sciences, São Paulo State University (Unesp), 14884-900 Jaboticabal, SP, Brazil; giovani.stingelin@unesp.br (G.M.S.); marina.mechler@unesp.br (M.L.M.-D.); gabriel.storino@unesp.br (G.Y.S.); karina.sonalio@unesp.br (K.S.); henri_almeida2003@yahoo.com.br (H.M.d.S.A.); fernando.petri@unesp.br (F.A.M.P.)

**Keywords:** enzootic pneumonia, piglets, respiratory diseases, metaphylaxis

## Abstract

*Mycoplasma hyopneumoniae* is the primary agent of Swine Enzootic Pneumonia (SEP). Vaccines reduce the clinical manifestation of the disease but do not prevent infection. The present study aimed to evaluate the use of antimicrobial drugs to minimize the impact of *M. hyopneumoniae*. For this, 32 pregnant female pigs and their litters were selected and then followed from birth to slaughter. The study involved three experimental groups that received metaphylactic treatment with different protocols involving tilmicosin, valnemulin, tulathromycin, and a control group to compare the effect of treatments against *M. hyopneumoniae* infection throughout the phases. Performance data were recorded, and the piglets were evaluated for the occurrence of cough. Nasal swab and blood collection was conducted periodically to detect *M. hyopneumoniae* shedding and anti-*M. hyopneumoniae* IgG, respectively. At slaughter, the lungs of animals from all groups were evaluated, and samples were collected for histopathological examination and qPCR for *M. hyopneumoniae* detection. All protocols promoted a reduction in consolidation lung lesions when compared to the control group. Individuals treated with valnemulin showed increased performance results, lower mortality, and low bacterial load in the lung. The results are promising and may indicate an alternative in the strategic control of *M. hyopneumoniae* infection in pigs.

## 1. Introduction

The occurrence of respiratory diseases in swine is multifactorial, and infectious pathogens, such as *Mycoplasma hyopneumoniae*, are commonly identified. It has been determined that Swine Enzootic Pneumonia (SEP) is caused primarily by *M. hyopneumoniae*, which causes immunosuppression and predisposes animals to infections by secondary pathogens such as *Pasteurella multocida* [1]. *Mycoplasma hyopneumoniae* infections are highly prevalent worldwide and result in financial losses for the swine industry. Economic losses due to respiratory diseases are associated with treatment costs, vaccination, decreased performance, and increased mortality [2,3].

Infection by *M. hyopneumoniae* starts with the adhesion of microorganisms to the ciliated epithelium of the respiratory tract of swine, promoting a disordered and chronic inflammatory response, accompanied by the modulation of the cellular and humoral immune response that maintains the pathogen for long periods in the infected animals [1]. In farms, the gilt is the principal source of *M. hyopneumoniae* infection, exposing the litter to the pathogen early during the farrowing phase. Piglets infected in subsequent phases will shed the pathogen and likely infect other susceptible individuals, a factor that is potentiated by mixing pigs [4].

Even though vaccines against *M. hyopneumoniae* reduce clinical signs and lung lesions, they do not prevent infection because the prevalence of infected individuals is greater than 70% in commercial herds, even with vaccination [3,5]. Concerning the partial effect of vaccines against *M. hyopneumoniae*, control strategies involving the application of appropriate drug protocols, with antimicrobial molecules of high efficacy against the pathogen, could reduce the transmission between sows and piglets at farrowing and between piglets in the nursery and finishing phases, minimizing the impact of SEP on commercial herds.

Among the drugs that do not belong to the Group III (Quinolones, Cephalosporins, Polymyxins, and Phosphonic Acids), a category of drugs classified by the World Health Organization (WHO) as a last resort for use in animals [6], valnemulin, tilmicosin and tulathromycin are the molecules that have been proven effective against respiratory pathogens in general and possibly in controlling EP [7,8,9,10]. Finally, given the search for alternatives in the control of *M. hyopneumoniae* that can be supported with vaccination and are in line with the rational use of antimicrobials, the objective of this study was to evaluate the effects of different drug protocols in the control of Enzootic Pneumonia in commercial pig production and to provide data on a metaphylactic drug treatment strategy with the antimicrobials tilmicosin, valnemulin, and tulathromycin against *M. hyopneumoniae* infection in pigs.

## 2. Results

### 2.1. Quantification of IgG in Swine Serum over Time

Serum samples from 12 individuals from each group were submitted to the ELISA test for detection and quantification of IgG anti-*M. hyopneumoniae* (Table 1 and Appendix A Appendix A). Differences between the mean values of S/*P* ± standard error (SE) were observed only at 63 days, where the mean value of G4 (0.599 ± 0.17) did not differ from the others, and the mean value recorded for G2 (0.997 ± 0.15) was higher than the mean values of G1 (0.188 ± 0.04) and G3 (0.183 ± 0.04), which did not differ from each other. At the other collection points, no statistical differences were observed. At 93 days of age, G2 had a numerically higher S/*P* value than the other groups with a significant trend compared to G1 and G3 (*p* = 0.06). At 123 days of age, G2 and G4 presented numerically higher values, and at 151 days, the mean S/*P* values of G2 were higher than the others (Figure 1).

When comparing the S/*P* values between repeated measures over time for each group, it was observed that for G1 at 24 days the antibody titer did not differ from the others. At 45 days, the antibody titer was lower than those at 123 and 151 days. At 151 days of age, there was no difference compared to 45 and 63 days, which did not differ from the other points (*p* = 1 × 10^−4^). For G2, the medians of S/*P* recorded at 123 and 151 days were higher than at 24 and 45 days. At 63 and 93 days, there were no differences between them or among other points (*p* = 2.8 × 10^−5^). In G3, differences were also observed between the values at 123 and 151 days compared to 24 and 45 days, while the other points did not differ from each other or from the others (*p* = 9.8 × 10^−5^). Finally, in G4, the S/*P* values at 123 and 151 days were higher than those at 24, 45, and 63 days, which did not differ, and at 93 days, no differences were observed when compared to the other time points.

At 45 days of age, no serum sample was positive for anti-*M. hyopneumoniae* IgG. In G2, no animal was positive until 63 days of age, the moment of the highest percentage of seroconversion for G2 at this time of collection when compared to the other groups.

### 2.2. Detection and Quantification of Mycoplasma hyopneumoniae by qPCR in Nasal Swab

Nasal swabs were collected from the eight sows of each group on the 7th day before farrowing (d-7) and on the farrowing day (d0). A G2 sow was positive in only one of the duplicates, most likely owing to the Monte Carlo effect [11]. A primiparous sow presented a positive result on the farrowing day. A sow from G3, second farrowing, was also positive on the 7th day before farrowing (d-7) (Monte Carlo effect). There was no statistical difference between the groups, although at 151 days there was a trend toward significance between G3 and G4 (*p* = 0.08).

In Appendix A Appendix A, it is possible to verify that the medians of the quantifications of *M. hyopneumoniae* in piglets were 0.00 for all the groups, and statistical differences were not observed over time for each time point (T1, *p* = 0.270; T2, *p* = 0.140; T3, NA; T4, *p* = 0.50; T5, *p* = 0.130; T6, *p* = 0.07). In Figure 1, it is possible to observe the percentage of positive individuals at overtime for different groups.

Concerning the repeated measures analysis over time, only G1 showed differences between the quantification values. For G1, the quantification values recorded at 123 days were higher than the values observed at 45, 63, and 93 days, which in turn did not differ from each other or from the quantification observed at 151 days.

When considering all time points, six animals from G1 were positive in at least one of them, and two of those animals were positive with the Monte Carlo effect [11]. In G2, five animals were positive at some point in all collected periods, and two of them were positive with the Monte Carlo effect. In G3, 12 animals were positive, in which four presented the Monte Carlo effect. In G4, nine animals were positive, where three of them also showed the Monte Carlo effect. There was no statistical difference among groups regarding the number of samples positive for *M. hyopneumoniae* or concerning quantification. Regarding positive samples from animals at different ages, at 63 days of age, none of the groups had positive animals. On the other hand, at 123 days of age, all groups had positive individuals, and the same occurred in G1, G2, and G3 at 151 days of age.

### 2.3. Assessment of Health Indicators

#### 2.3.1. Cough Examination

Periodically throughout the study, cough counts were performed to calculate the cough index. Up to 63 days of age, no occurrence of cough was observed in any of the groups. At 93 days, no cough was recorded in G1 and G3, while in G2 the cough index was 0.39% and in G4 was 1.15%. At 123 days of age, the cough indexes were: 1.98% in G1, 0.39% in G2, 1.18% in G3, and 5.75% in G4. At 151 days, the cough index was 2.03% in G1, 0.78% in G2, 0.80% in G3, and 1.16% in G4. Despite the numerical differences, there was no significant difference between the groups at the level of *p* < 0.05 by the non-parametric Kruskal–Wallis test.

#### 2.3.2. Mortality

Regarding the percentage of mortality in the nursery, only one animal died in G1 (1.11%), and no other deaths were recorded in the other groups. There was no statistical difference between the groups in this period. In the finishing phase, six animals from G1 (6.74%), two animals from G2 (2.3%), seven animals from G3 (7.69%), and one animal from G4 (1.15%) died.

In G1, four of the six animals that died had clinical respiratory signs, and two of them had clinical signs suggestive of meningitis. In G2, one animal died from intestinal torsion and the other by wasting, showing pulmonary involvement and a pattern of an interstitial lesion in the parenchyma, characteristic of influenza infection. In G3, two of the seven animals that died had clinical signs of meningitis, the other two had clinical signs of pneumonia with pulmonary involvement at necropsy, and another three animals died from non-infectious causes. Finally, in G4, only one animal died at the finishing phase with clinical signs of pneumonia and pulmonary involvement observed at necropsy.

If we consider the total mortality from weaning to slaughter, mortality was numerically higher in G1 (7.78%) and G3 (7.69%) compared to G2 (2.3%) and G4 (1.15%). However, no significant difference was noted between the groups, according to the confidence interval of the Wilson method.

### 2.4. Evaluation of Performance Indicators

At weaning, G1 consisted of 91 piglets, G2 of 87 piglets, G3 of 91 piglets, and G4 of 87 piglets. The weight averages of the different groups over time from T1 to T6 (151 days of age for G1, G2 and G3 pigs and 145 days of age for G4 pigs) are shown in Table 2.

At weaning, piglets from G2 were heavier (kg ± EP) (7.05 ± 0.17) when compared to the other groups (*p* < 0.001), which did not differ among themselves. Upon leaving the nursery at 63 days, individuals from G2 (22.42 ± 0.40) and G4 (23.35 ± 0.43) were significantly heavier than those from G1 (20.64 ± 0.42) (*p* = 0.02 and *p* < 0.001) and G3 (20.66 ± 0.48) (*p* = 0.02 and *p* < 0.001), which did not differ from each other. Concerning slaughter weight, regardless of age, G1, G2, and G3 were slaughtered at 151 days of age and G4 at 145 days of age. At slaughter, G2 (113.33 ± 1.28) was heavier than the other groups (G2 × G1, *p* = 0.04; G2 × G3, *p* = 0.01 and G2 × G4, *p* < 0.001), which did not differ from each other.

Considering that the ADWG of G4 was 0.974 kg in the period from 63 to 145 days of age, at 151 days of age, G2 (113.33 ± 1.28) was statistically heavier than G3 (104.95 ± 1.47), but there was no difference in weight for G1 (107.18 ± 1.50) and G4 (108.29 ± 2.34), which in turn would not differ from the others (*p* < 0.05).

Regarding the weight distribution within each group at slaughter, G2 presented a more uniform normal distribution curve with a slight deviation to the right, indicating higher influent values (Figure 2). The coefficients of variation regarding the weight of animals at slaughter were 14% for G1, 11% for G2, 13.5% for G3, and 13% for G4.

Regarding the assessment of average daily weight gain (ADWG), in Table 3 it is possible to verify the performance of the different groups in the nursery, finishing phase and from 25 days (weaning) to slaughter, at 151 days of age for G1, G2 and G3, and 145 days of age for G4.

In the nursery, the ADWG (g ± SE) of G4 (0.45 ± 0.009) was higher than those of the other groups, which did not differ from each other. At termination, the ADWG of G2 (1.07 ± 0.013) was superior to those of the other groups, which did not differ from each other. The ADWG from weaning to slaughter for G2 (0.87 ± 0.01) was higher than those for G1 (0.82 ± 0.012) (*p* = 0.01) and G3 (0.81 ± 0.011) (*p* = 0.03), and there was no difference for G4 (0.84 ± 0.012).

In the evaluation of feed conversion in the nursery period (g ± SE), G4 (1.69 ± 0.042) and G2 (1.76 ± 0.051) had the best feed conversion with statistical difference when compared to G1 (2.01 ± 0.047) (*p* < 0.001 and *p* = 0.006) and G3 (2.11 ± 0.63) (*p* < 0.001 and *p* < 0.001), which did not differ from each other. In the assessment of finishing feed conversion, G2 (2.02 ± 0.028) was better than G1 (2.12 ± 0.034) (*p* = 0.03), and there was no statistical difference for G3 (2.12 ± 0.031) and G4 (2.03 ± 0.033). In the evaluation of feed conversion of the herd from weaning to slaughter, G2 (1.97 ± 0.025) and G4 (1.97 ± 0.031) had better feed conversion rates when compared to G1 (2.12 ± 0.033) (*p* = 0.001 and *p* = 0.03) and G3 (2.14 ± 0.031) (*p* < 0.001 and *p* = 0.01), and the latter two did not differ from each other (Table 4).

### 2.5. Mycoplasma hyopneumoniae Quantification in Bronchoalveolar Lavage Fluid and Lung Tissue Samples by qPCR

Lung fragments and BALF samples from 15 individuals from each group were subjected to DNA extraction and qPCR for detection and quantification of *M. hyopneumoniae* (Table 5 and Table 6). The average quantification values recorded in lung tissue samples ± standard error (SE) of G2 (6.74 × 10^4^ ± 2.90 × 10^4^ copies/µL) did not differ from the others, while G1 (1.34 × 10^5^ ± 4.84 × 10^4^ copies/µL) showed a mean higher than those of G3 (3.15 × 10^4^ ± 9.72 × 10^3^ copies/µL) and G4 (2.41 × 10^4^ ± 1.06 × 10^4^ copies/µL) (*p* = 0.02), which did not differ from each other. Regarding the quantification of *M. hyopneumoniae* DNA in BALF samples, no significant differences were observed between the groups. The association between quantification values for *M. hyopneumoniae* in the lung and BALF were analyzed for each group separately by the Kendall correlation coefficient, and only for G1 was a significant association observed (*p* < 0.05, Tau = 0.47).

### 2.6. Lung Injuries Examination

#### Determination of the Pneumonia Index

The lungs of the individuals in each group were individually evaluated, recording the degree of consolidation lesions of each lobe so that the total affected lung area could be calculated (Appendix A Appendix A). Concerning the mean of pulmonary consolidation for G1 (9.82 ± 2.02), G2 (6.92 ± 1.31), G3 (7.47 ± 1.57) and G4 (7.69 ± 1.27), no significant difference was observed among the groups (*p* = 0.60).

The prevalence of pulmonary consolidation in G1 was 96.55% (82.82 to 99.39%), and the pneumonia index was 1.52. For G2, the prevalence of pulmonary consolidation was 91.3% (73.2 to 97.58%), and the pneumonia index was 1.17. For G3, the prevalence of pulmonary consolidation was 80% (62.69 to 90.5%), and the pneumonia index was 1.20. In G4, the prevalence of total lung consolidation was 93.33% (78.68 to 98.15%) while the pneumonia index was 1.27 (Appendix A Appendix A).

### 2.7. Histopathological Evaluation

Fragments of 15 lungs were collected in each group from the areas between lesions and healthy tissue and were routinely processed with hematoxylin/eosin staining. The slides were evaluated under a light microscope, and the degree of microscopic lesions was classified according to the methodology of Casalmiglia et al. [12]. The lungs sampled were collected from the same individuals who underwent periodic serum collections throughout the study (Appendix A Appendix A).

Differences were not observed regarding the median of the degree of injury for G1 (3.00 ± 2.60), G2 (3.00 ± 2.13), G3 (2.50 ± 1.71), and G4 (3.00 ± 2.83) (*p* = 0.13) (Appendix A Appendix A). Numerically, G1 and G4 had the highest number of animals in classification 3, with 11 lungs in each group, and were also the groups with the lowest absolute number of lungs classified with grades below 2. G3 was the group with the highest number of lungs without any type of injury (grade 0) (Figure 3).

## 3. Discussion

In the present study, to clarify the role of metaphylactic treatment with molecules recognized as effective in respiratory disease treatment and the control of *M. hyopneumoniae*, laboratory analysis and productivity evaluations were conducted in animals submitted to different drug protocols in commercial swine production. Concerning the development of humoral immunity evaluated through the detection and quantification of IgG anti-*M. hyopneumoniae*, the results showed evidence that antibody titers for G2 were higher than in the other groups. Regarding the quantification of *M. hyopneumoniae* in the nasal swab and the cough examination and mortality, no significant differences were observed among the groups. Regarding the performance indicators, better results for the body weight were evidenced in G2 at all points evaluated. Similar results were observed concerning ADWG and feed conversion, with superior results in G2 and G4, both groups being medicated with valnemulin.

Our results indicated that only 6.25% (2/32) of the sows were shedding *M. hyopneumoniae* in the days before farrowing, despite the history of high prevalence of the pathogen in the farm where the study was conducted. Concerning the intermittent pattern of bacterial shedding in infected individuals, there is a possibility that the number of positive sows was underestimated [13]. A similar evaluation conducted in a vaccinated herd found that only 2.3% of the females were shedding *M. hyopneumoniae* at the time of sampling [14].

In breeding herds, medication with tivalosin, a semi-synthetic macrolide, before farrowing showed promising results in reducing the prevalence of *M. hyopneumoniae* [15]. In the present study, no significant differences were observed among groups regarding *M. hyopneumoniae* quantification in nasal swab samples or the prevalence of positive individuals. However, despite the nasal swab being a good indicator of *M. hyopneumoniae* excretion, it has limitations when used to determine the prevalence of positive individuals in the herd owing to the intermittent excretion pattern [16].

At weaning, only in G2 were there no positive individuals for *M. hyopneumoniae* in nasal swab samples, while in the other groups, the prevalence ranged from 8 to 25%. As described by Fano et al. (2007), the percentage of positive individuals at weaning was positively correlated with lung lesions at slaughter [17]. In the present study, it was observed that the rates of lung area with consolidation lesions ranged from 91.30% (G2) to 96.55% (G1), while the pneumonia index ranged from 1.17 (G2) to 1.52 (G1). Although the correlation between excretion data and lung lesions at slaughter was not observed, the lower numerical values observed in G2 are possible indicators that the impact of SEP was lower in G2, which was confirmed by the results of the performance indices.

In farms with animals infected by *M. hyopneumoniae*, at close to 120 days of age a peak of clinical signs was reported, evidenced by the occurrence of cough in infected individuals [13]. In the present study, at 123 days the cough index varied between 0.39% in G2 and 5.75% in G4. However, despite the remarkable numerical difference, a significant difference based on the confidence interval was not observed, probably because of the sample size. Moreover, at 123 days the ratio of positive individuals for *M. hyopneumoniae* among the groups ranged from 9% in G2 to 33% in G1. The result possibly indicates that, directly or indirectly, the occurrence of cough and the prevalence of positive individuals may be associated and that the impact of SEP actually differed among groups.

Compared with other field studies, the incidence of cough was low for all study groups [18]. Only after 93 days of age was it possible to evidence the occurrence of cough in all groups. When analyzing the data from positive samples and quantification of *M. hyopneumoniae* in the nasal swab samples, as well as the cough index, we observed that the data, based on their characteristics and descriptive statistics, did not meet the assumptions for the application of parametric tests, which considerably reduced the chances of detecting significance among treatments.

Our findings indicated a considerable increase in the prevalence of positive animals at 123 days of age in G1 (33%) and G3 (33%), and at 151 days of age in G4 (25%). The G2 was the only group where positivity did not increase considerably and reached a maximum of 17% at 93 days of age, probably owing to the strategic treatment with a therapeutic dose of valnemulin at the beginning of the finishing phase, from 64 to 78 days of age, which reduced the infection pressure and transmission of the pathogen among the animals in the group. In farms where antibiotics were withdrawn during the finishing phase, up to 95.60% of animals tested positive for *Mycoplasma* spp. [19].

Still, regarding the quantification of *M. hyopneumoniae* in the lung tissue, G1 presented a higher quantification value when compared to G3 and G4. G3 and G4 did not differ between each other or from G2. Regarding the quantification of *M. hyopneumoniae* in BALF, there was no significant difference among the groups. Models to analyze the correlation between the quantification of *M. hyopneumoniae* in BALF and lung were carried out, resulting in a significant direct correlation between the variables (*p* = 0.01 and Tau = 0.46) only for G1.

Regarding the histopathological lesions, three lungs from G1 (20%), one lung from G2 (6.66%), one lung from G3 (7.14%), and one lung from G4 (8.33%) showed lesions suggestive of SEP, reiterating that the treated groups had a lower incidence of co-infections, possibly because of the more efficient control of *M. hyopneumoniae*.

Piglet seropositivity prevalence was reported to be 8% in G1 and G3 and 25% in G4, where the animals were positive until 24 days of life, probably because of passive immunity that can last up to 28 days. At 45 days of age, after two doses of vaccine for *M. hyopneumoniae*, none of the individuals in groups G1, G2, and G3 had antibody levels above the detection threshold, and 14% of the piglets in G4 were positive for the presence of anti-*M. hyopneumoniae* IgG, which may be associated with the onset of seroconversion. This agreed with Wilson et al. [20], who reported that maternal antibodies could be present up to the fourth week of life in piglets.

At 45 days of age, an absence of antibodies was observed in all groups, and from 63 days of age, seropositive animals were observed in all groups, evidencing a possible effect of vaccination on individuals after the second dose. When comparing the seroprevalence profile over time among groups, it was observed that G1, G3, and G4 presented similar patterns, with up to 50% of individuals having anti-*M. hyopneumoniae* IgG levels and a gradual increase in the percentage up to 150 days, while in G2, 91.66% of the individuals were positive at 63 days, and the proportions remained high until slaughter. Therefore, the effective action of valnemulin against respiratory pathogens may result in a general improvement in respiratory health, contributing to a better immune response in G2 animals.

There was no evidence that levels of anti-*M. hyopneumoniae* were associated with a protective effect against infection [21]. Studies have shown beneficial results concerning vaccination of breeding females, although it is not possible to determine whether the positive results on piglets are due to maternal antibodies or specific lymphocytes against the pathogen transferred by colostrum [22]. Concerning the fact that only G2 did not present seropositive individuals at 24 days of life, it is possible that valnemulin medication in sows was effective in controlling the infection by *M. hyopneumoniae,* which consequently led to an absence of seroconversion by these individuals, leading to non-transfer of antibodies to piglets through colostrum.

Studies in pigs have reported the potential of macrolides in modulating the immune response, altering the concentrations of defense cells, and regulating the release of cytokines [23]. A similar profile was also observed for valnemulin, which presented a modulatory effect on lipopolysaccharide-stimulated macrophages [9]. In the present study, however, because the cytokine profiles were not evaluated and the study was not conducted in a controlled environment, it was not possible to demonstrate that the antibiotics modulated the immune response. Despite this, evidence of differences among treatments was observed, especially when considering the performance of individuals.

At the beginning of the growing period, G3 received tilmicosin, which has a broad spectrum of action for respiratory pathogens. However, the ability of tilmicosin to penetrate the blood–brain barrier is low [24], and the MIC for *Streptococcus suis* is high, greater than >128 μg/mL [25]. This molecule does not have a pharmacokinetic spectrum of action against *S. suis*, however, which may explain the high mortality from meningitis in individuals in this group. Metaphylactic medication with valnemulin at the beginning of the growing period in G2 and G4 resulted in the absence of streptococcal meningitis in piglets during the growing and finishing phases.

In the nursery, G4 presented the best performance, with an ADWG of 0.450 kg, statistically superior to G1, G2, and G3, which did not differ from each other. Concerning the good performance of G4 in the nursery, the weight at 63 days of age resulted in no significant difference in relation to G2 (22.42 kg) at the level of *p* < 0.05. Although weighings were not performed at 145 days, it is possible that G4, based on the finishing ADWG, had superior values to the other groups. Unfortunately, the slaughter of G4 needed to be earlier than the others, at 145 days of age, owing to slaughterhouse logistics. If we considered the average weight gain during the growth and finishing phase for G4 and estimated the weight at 151 days, the results would indicate that G2 is statistically superior to G3 but not superior to G1 and G4, which did not differ from each other, at the level of *p* < 0.05.

Concerning the ADWG from 63 to 151 days of age, G2 (1.070 kg) was more efficient and performed significantly better than G1 (1.010 kg), G3 (0.99 kg), and G4 (1.010 kg), which did not differ from each other. However, regarding the daily weight gain from weaning to slaughter age, G2 (0.870 kg) was significantly higher than G1 (0.820 kg) and G3 (0.810 kg) and did not differ from G4 (0.840 kg), which in turn did not differ from G1 and G3. Although in this scenario, the early slaughter of individuals from G4 may have compromised the results of average weight gain, the results from G2 and G4 indicated that the groups may have had better results because of the common antimicrobial, valnemulin. Herds positive for *M. hyopneumoniae* and with high infection pressure had a worse feed conversion by 0.080 kg when compared to free herds [4]. Groups 2 and 4 were not free of *M. hyopneumoniae*; however, from a productivity perspective, they showed promising results.

Regarding feed conversion, in the nursery G2 (1.760 kg) and G4 (1.690 kg) were significantly better than the others, while in the finishing phase, from 63 days of age to slaughter, G2 had statistically better feed conversion (1.970 kg) than G1 (2.120 kg) although it did not differ significantly from G3 (2.120 kg) and G4 (2.030 kg). In addition, G1, G3, and G4 did not differ from each other. If we considered the feed conversion from weaning to slaughter, G2 (1.970 kg) and G4 (1.970 kg) were more efficient groups than G1 (2.120 kg) and G3 (2.140 kg) at *p* < 0.05 by Tukey’s test.

An important observation of this study was that in G1, minimum levels of antimicrobials were maintained, relying only on molecules intended to control enteric diseases. However, because of the meningitis outbreaks, we needed to give a new pulse of amoxicillin, increasing the total amount of antimicrobials administered to individuals in G1, totaling 340 mg/kg.

In G3, 178 mg of antimicrobial was administered per kg of slaughtered swine in total. In the drug protocol of this group, at the beginning of the growing phase, a pulse of tilmicosin was administered, and this molecule exhibited no spectrum of action against *S. suis*. Even though the farm had no history of streptococcal meningitis, animals in G3 also died from meningitis. Finally, G3 also received another pulse of amoxicillin at 437 mg of antimicrobial per kg of slaughtered swine.

In G2, 116 mg of antimicrobial per kg of slaughtered swine was administered, and in G4, the animals received 102 mg of antimicrobial per kg of slaughtered swine.

Pleuromutilins have shown superior action to macrolides against *Mycoplasma* sp. Within this group of antimicrobials, valnemulin is described as 30 times more effective than tiamulin against *Mycoplasma* sp. in vitro [26].

When administered in feed to swine, valnemulin reaches peak plasma concentration within 4 h and has high bioavailability in the lung [27,28], where it is proven to be effective against *Mycoplasma* [29]. Although there are few studies evaluating the effectiveness of valnemulin in the control of respiratory diseases, when considering the molecule characteristics, it has been demonstrated that possible favorable results can be achieved when using the molecule in strategic drug protocols aiming at controlling *M. hyopneumoniae* infection and, consequently, the SEP. However, regarding both macrolides and pleuromutilins, we lack studies to prove the effectiveness of the drugs against the complex of respiratory diseases, which includes *M. hyopneumoniae* infection.

Pleuromutilins and macrolides, despite having similar mechanisms of action, are distinct from the pharmacodynamics point of view. Thus, we observed in the present study that although G2, G3, and G4 were all treated with valnemulin, there were differences between the protocols that promoted different results between the groups. There was evidence in the present study that valnemulin may be more effective than tulathromycin in the treatment of sows, possibly because of the rapid lung bioavailability of the molecule and the greater effectiveness of pleuromutilins compared to macrolides in controlling *M. hyopneumoniae*, as previously described. Only G2 and G4 were treated with valnemulin while G3 was treated with tulathromycin, which belongs to the macrolide class, and protocols based only on valnemulin at different doses (G2 and G4) may produce similar results from the point of view of zootechnical indicators.

Macrolides and pleuromutilins are two of the few classes of antimicrobial molecules that can be used in veterinary medicine and that are, at least in vitro, proven to be effective in controlling *M. hyopneumoniae*. Within this aspect, the present study explored protocols based only on molecules belonging to these classes. In addition, valnemulin was used in at least one of the phases in the three groups. Although there is evidence that the results of G2 and G4, involving exclusively valnemulin administration, were superior compared to G3, it is not possible to affirm that the results are related only to valnemulin, which is a limitation of the present study. Differences in the treatment of the sows before farrowing and the piglets during the growth phase in G3 may have influenced the differences in the results between the groups, mainly in the performance indices. Future studies with drug protocols based on molecules that are more distinct from each other will possibly corroborate the conclusions of the present study and attest to the positive effect of valnemulin in the control of SEP.

It is necessary to emphasize that field studies involve numerous variables that are often impossible to measure or categorize; thus, establishing associations is challenging, especially when only two factors are considered. Associations are the only alternative when assumptions about more robust analyses are not met. 

Based on the data obtained from the present study, it was possible to conclude that the strategic use of antimicrobials alone may not be enough to control SEP, although it presents favorable results from the productive point of view. Furthermore, the reduction in the amount of medication administered, the maintenance, and even the productivity improvement are achievable results when using highly effective antimicrobial molecules. Nevertheless, it is essential to emphasize that sanitary measures must be based on technical and scientific data so that maximum effectiveness in the control and possibly the eradication of respiratory pathogens is achieved.

## 4. Material and Methods

### 4.1. Study Farm

The present study was carried out in a commercial farrow-to-finish operation farm of 450 sows located in the municipality of Amparo da Serra, State of Minas Gerais, Brazil, having a history of respiratory disease in nursery pigs (21 to 65 days of life) and finishing pigs (70 to 150 days of life). At the farm, sows and piglets were vaccinated against *M. hyopneumoniae* using the Respisure vaccine (Zoetis Laboratory, Louisville, KY, USA). In addition, the females received the Farrowsure B Gold vaccine (Zoetis Laboratory, Louisville, KY, USA) against swine parvovirus, *Erysipelothrix rhusiopathiae* and *Leptospira* spp., the Litterguard LT-C vaccine (Zoetis Laboratory, Louisville, KY, USA) against *E. coli* and *Clostridium perfringens* type C, and the ARadicator vaccine (Zoetis Laboratory, Louisville, KY, USA) with *Pasteurella* toxoid *multocida* type D toxoid, *Pasteurella* inactivated type D *multocida* and *Bordetella* inactivated *bronchiseptica*. The piglets were also vaccinated against porcine circovirus type 2 (PCV2) with the Ingelvac Circoflex vaccine (Boehringer Ingelheim Laboratory, Ingelheim am Rhein, Germany) and against *Glaesserella parasuis* and *Pasteurella multocida* using an autogenous vaccine produced by Ipeve Laboratory, Belo Horizonte, MG–Brazil. All procedures of this study were approved by the Ethics Committee for the Use of Animals (CEUA), Faculty of Agrarian and Veterinary Sciences, Unesp Campus of Jaboticabal–SP, under protocol n° 009149/18.

### 4.2. Experimental Design

A completely randomized design was used to form four groups of eight females each (G1-G4), or 32 females in total. In addition to the sows and gilts, the groups were composed of their respective litters, which were evaluated from birth to slaughter. The piglets were submitted to different antimicrobial protocols and periodically evaluated. Each breeding group consisted of two gilts (nulliparous) and six sows of different parities, varying from 2 to 9, and their litter. Females from different groups were housed in separate rooms, and individuals from the same group were kept in the same barn.

The piglets were identified at birth with earrings, and on the 3rd day of life they were weighed and separated into the different groups. Piglets weighing more than 3.3 kg and less than 1.4 kg at 3 days of age were removed from the study to standardize and obtain approximate weight averages between groups. Feed was provided twice daily and met the nutritional requirements for pigs [30], and water was provided ad libitum.

The groups composed of females and their litters were submitted to different drug protocols (Figure 4). Group 1 (G1) was considered a control group as it was submitted to a minimal treatment protocol when compared to the protocol regularly adopted on the property, aiming only at the maintenance of production rates and the prevention of enteric diseases. The other groups, in addition to being submitted to the protocol described for G1, were submitted to an additional drug protocol.

The four groups were characterized as follows: Group 1 (G1) comprised eight females not medicated before farrowing. Piglets from G1 (*n* = 89) received amoxicillin in feed at a dose of 15 mg/kg body weight (FARMAXILIN^®^, Farmabase Saúde Animal, Jaguariuna, Brazil), and 120 ppm of halquinol (H-MAX^®^, Farmabase Saúde Animal, Jaguariuna, Brazil) during the first 14 days of the nursery. In the growing and finishing phase, they received feed with the performance enhancer enramycin (Enramax^®^, Farmabase Saúde Animal, Jaguariuna, Brazil); Group 2 (G2): eight females medicated in feed with valnemulin (Rovax^®^, Farmabase Saúde Animal, Jaguariuna, Brazil) at a dose of 10 mg/kg during the 7 days prior to parturition. Piglets from G2 (*n* = 90) received 10 mg/kg of valnemulin, 15 mg/kg of amoxicillin (FARMAXILIN^®^, Farmabase Saúde Animal, Jaguariuna, Brazil) and 120 ppm of halquinol in feed (H-MAX^®^, Farmabase Saúde Animal, Jaguariuna, Brazil) in the first 14 days of the nursery, and at the beginning of the growing phase the animals received 10 mg/kg valnemulin for another 14 days; Group 3 (G3): eight females, medicated by the intramuscular route with tulathromycin in a single dose of 2.5 mg/kg, 7 days before parturition. Piglets from G3 (*n* = 94) were treated with 10 mg/kg of valnemulin, 15 mg/kg of amoxicillin (FARMAXILIN^®^, Farmabase Saúde Animal, Jaguariuna, Brazil), and 120 ppm of halquinol in feed (H-MAX^®^, Farmabase Saúde Animal, Jaguariuna, Brazil) in the first 14 days of nursery. At the beginning of the growing phase for 14 days, they received 20 mg/kg of tilmicosin (HI-BAC^®^, Farmabase Saúde Animal, Jaguariuna, Brazil) in feed; Group 4 (G4): eight females were medicated in feed with valnemulin (Rovax^®^, Farmabase Saúde Animal, Jaguariuna, Brazil) at a dose of 5 mg/kg of body weight for 7 days before parturition. Piglets from G4 (*n* = 89) received 5 mg/kg of valnemulin, plus 15 mg/kg of amoxicillin (FARMAXILIN^®^, Farmabase Saúde Animal, Jaguariuna, Brazil) and 120 ppm of halquinol in feed (H-MAX^®^, Farmabase Saúde Animal, Jaguariuna, Brazil) in the first 14 days of nursery. They received another 14 days of valnemulin at a dose of 5 mg/kg in the growing phase. G2, G3, and G4, as well as the control group, received the performance enhancer enramycin (Enramax^®^, Farmabase Saúde Animal, Jaguariuna, Brazil) in the growing and finishing diets (from 79 days of age to slaughter) (Figure 4).

### 4.3. Collection of Blood Serum Samples

Of the total number of piglets born from each group of sows, at weaning (24 days), 12 individuals were chosen randomly to undergo a clinical evaluation and blood collection every 21 days until 151 days of age, totaling six collection time points. Blood samples were collected by puncturing the jugular vein after antisepsis with 70% alcohol, using 25 × 0.8 mm needles and vacuum collection tubes (BD Vacutainer^®^, Franklin Lakes, NJ, USA).

Tubes containing the blood were centrifuged at 1500× *g* for 10 min (Centrifuge 5804 R, Eppendorf^®^, Hamburger, Germany) immediately after collection. The volume of serum obtained was aliquoted, in duplicates, into 2 mL sterile plastic microtubes, free of RNAse and DNAse, and stored at −20 °C until further processing.

Swab samples were collected from each of the 12 animals throughout the study period every 21 days to assess *M. hyopneumoniae* excretion. For sampling, the animals were restrained using a snare, and the swabs were inserted into the nasal cavity through the orifice of both nostrils, performing light rubbing of the swab on the nasal mucosa. Swabs were collected in duplicate and stored in 2 mL sterile plastic microtubes, free of RNAse and DNAse containing PBS (pH 7.4) and stored at −80 °C until processing for qPCR.

### 4.4. Cough Index

At each sample collection time point, before physical restraint, the animals were submitted to the cough evaluation to estimate the cough index, as described by Soncini and Madureira (1998) [31]. The methodology consisted of stimulating the movement of individuals within the pen perimeter for 1 min, followed by 1 min of rest. During rest, the occurrence of cough was recorded collectively by the group. The process was repeated two more times, and the average of the three counts formed the following equation:Cough frequency (%) = (Average of the three counts/number of animals in the pen) × 100.

### 4.5. Zootechnical Performance

Feed consumption was determined daily by weighing the leftovers after 24 h. Feed conversion was estimated for individuals throughout the phases based on feed consumption and weight gain [32]. Average daily weight gain (ADWG) was calculated based on periodic weighing at the beginning and end of the production phases.

### 4.6. Mortality

The mortality of individuals was evaluated during nursey (from 25 to 63 days of age), finishing (from 64 days of age to slaughter), and from weaning to slaughter (25 days of age to slaughter). The suspected cause of death was also reported based on the history, clinical signs, and, if necessary, necroscopic examination.

### 4.7. Slaughter of Animals, Lung Evaluation, and Sample Collection

At the end of the finishing phase, all the piglets were slaughtered in a slaughterhouse according to the methods established by ordinance No. 711, 11/1/1995, of the Ministry of Agriculture, Livestock and Food Supply [33]. After evisceration, the respiratory set (trachea and lung) of 29 animals from G1, 23 animals from G2, 30 animals from G3, and 30 animals from G4, was identified and separated at the slaughter line.

The lungs of all pigs in each group were macroscopically evaluated, for the presence of lesions, scoring the degree of consolidation and extension of lesions. Each lobe was evaluated individually to estimate the percentage of the affected area, establishing a score from 0 to 4, with 0 assigned to lobes without evidence of injury; 1: from 1 to 25% injury; 2: from 26 to 50%; 3: from 51 to 75%; and 4: from 76 to 100%, as described in the literature [17].

The lung lesion score for each lobe was multiplied by the relative lung weight to calculate the total area injured. As the lobes did not represent equal parts of the total lung volume, the following relative percentages were assigned: right apical lobe = 11%, right cardiac lobe = 11%, right diaphragmatic lobe = 34%, left apical lobe = 6%, left cardiac lobe = 6%, left diaphragmatic lobe = 27%, intermediate lobe = 5% [34]. The degree of injury was assessed according to the total area of pneumonia, using the mean of each lobe score concerning the total lung area to obtain the pneumonia index [34,35,36]. Groups with a pneumonia index mean of up to 0.55 were considered pneumonia-free (Grade 0). The animals with a mean pneumonia index between 0.56 and 0.89 obtained an intermediate classification (Grade 1), in which the presence of pneumonia occurred. Animals with a pneumonia index above 0.90 were considered severely affected, with high levels of pneumonia (Grade 2).

After macroscopic inspection of the lungs, bronchoalveolar lavage fluid (BALF) was collected. For collection, sterilized scalpel blades were used to perform cross-sections at 2 cm from the bifurcation of the trachea. Directly into the bronchi, 20 mL of PBS (1×, pH 7.4; Sigma-Aldrich, Darmstadt, Germany) was dispensed, followed by a gentle massage through the lung parenchyma and aspiration with a sterile plastic serological pipette of 50 mL (Eppendorf, Hamburger, Germany), recovering between 10 mL of lavage. The recovered content was transferred to 50 mL falcon tubes (Corning, Corning, NY, USA) and later aliquoted into 2 mL cryotubes free of DNAse and RNAse (Corning, Corning, NY, USA), which were then stored in a freezer at −80 °C until further processing.

Fragments of lung tissue were obtained from the transition region between the healthy and affected tissue of the apical lobes. Smaller portions were collected with the aid of scalpel blades and sterilized tweezers and placed into 2 mL sterile plastic microtubes free of DNAse and RNAse (Eppendorf, Hamburger, Germany), intended for qPCR. Larger fragments of approximately 1 cm^3^ were collected and stored in 80 mL universal collection flasks containing 10% buffered formalin for subsequent histopathological analysis. Samples were immediately stored on dry ice and later transferred to a −80 °C freezer.

Lung fragments destined for histopathological analysis fixed in 10% buffered formalin were processed for hematoxylin/eosin staining. The slides were evaluated under a light microscope, and the microscopic lesions in the tissues were classified into five different degrees [20], as follows: 0 = absence of lesion; 1 = lesions of interstitial pneumonia and or catarrhal bronchopneumonia; 2 = mild or moderate infiltration of neutrophils, macrophages and lymphocytes in the airways and alveoli; 3 = perivascular or peribronchiolar lymphoplasmacytic hyperplasia, type II pneumocyte hyperplasia and presence of edema in the alveoli; and 4 = the same lesions as 3, plus the presence of perivascular and peribronchiolar lymphoid nodes.

### 4.8. Detection and Quantification of Antibodies by ELISA

The detection and quantification of anti-*M. hyopneumoniae* IgG in serum was performed with enzyme immunoassay. The commercial kit *M.hyo* Ab test (Idexx, Westbrook, ME, USA) was used, following the manufacturer’s guidelines. The plates were read in an absorbance microplate reader (iMark, Bio-Rad Laboratories Inc., California, USA) under a wavelength of 650 nm. The average optical densities (ODs) for each of the test samples were related to the ODs of the negative and positive controls (CN $\bar{x}$ − CP $\bar{x}$) to calculate the S/*P* values (sample/positive ratio) according to the formula: S/*P* = (ODs − CN $\bar{x}$) ⁄ (CP $\bar{x}$ − CN $\bar{x}$). Serum samples were considered positive if S/*P* > 0.3.

### 4.9. DNA Extraction and cPCR Performance for the Gapdh Gene

DNA extraction was performed using the in-house Tris-HCl protocol [37]. For BALF samples, centrifugation was performed (Centrifuge 5804 R, Eppendorf^®^, Hamburger, Germany) at 13,000× *g* at 4 °C for 20 min before the DNA extraction [21]. For the lung samples, 0.05 g of lung tissue was used. After DNA extraction, samples were stored at −20 °C until analysis with qPCR. Measurement of the DNA concentration of the samples was carried out with spectrophotometry, using the ThermoScientific NanoDrop™ One Spectrophotometer (Thermo Fisher Scientific^®^, Wilmington, NC, USA), having as an exclusion factor the samples that did not reach the purity of 1.8 to 2.0 in the 260/280 ratio. To rule out the presence of inhibitors in the extracted DNA samples and the occurrence of false negatives in qPCR for *M. hyopneumoniae*, all samples were subjected to a conventional PCR [38] targeting the endogenous gene Glyceraldehyde-3-phosphate dehydrogenase (*gapdh*). The 437 bp *gapdh* gene-amplified products were detected in horizontal agarose gel electrophoresis.

### 4.10. Detection and Quantification of M. hyopneumoniae by qPCR

Lung fragments, nasal swabs and BALF samples were tested and quantified by qPCR [39]. All DNA samples were tested in duplicates and the qPCR reaction was optimized from a previously published protocol, as described by Almeida et al. [40]. The primer pairs used in the reaction were based on the *M. hyopneumoniae* adhesion protein *p102* gene sequence (F-5′-AAGGGTCAAAGTCAAAGTC-3′; R-5′- AAATTAAAAGCTGTTCAAATGC-3′; and hydrolysis probe 5′-FAM-AACCAGTTTCCACTTCATCGCC-§BHQ2-3′).

Results were accepted only for those with standard deviation less than or equal to 0.5 cycle, and quantification data were used only if the efficiency obtained was between 90 and 105%; otherwise, samples were retested in triplicate. As a negative control in the qPCR reactions, sterile ultrapure water was used (Nuclease-Free Water, Promega^®^, Madison, Wisconsin, NC, USA) q.s.p. 10 µL. The standard quantification curve was performed with serial 10-fold dilutions (starting at 10^7^ to 10^1^ copies/μL) of synthetic DNA (GBlock^®^, IDT, Iowa City, IA, USA) containing the 150 bp fragment amplified by the pair of primers used in qPCR. The cycles were performed in a CFX-96 real-time thermocycler (Bio-Rad, Hercules, CA, USA) at the following settings: an initial denaturation cycle at 95 °C for 3 min, followed by 39 cycles at 95 °C for 15 s and a cycle of extension at 55.7 °C for 1 min.

## 5. Statistical Analysis

Quantitative and qualitative variables were analyzed to evaluate the treatment effects on the dynamics of infection, performance, and respiratory health. *Mycoplasma. hyopneumoniae* quantification in lung and BALF, IgG anti-*M. hyopneumoniae*, body weight, feed conversion, and respiratory health assessment data (cough, pneumonia index, and lung consolidation lesions) were analyzed with parametric and non-parametric statistical tests. Pearson’s (parametric) or Spearman’s (non-parametric) correlation (*p* < 0.05) analyses were performed to verify the association between the variables.

Finally, differences between the proportions of positive samples in qPCR for *M. hyopneumoniae* and the proportion of positive samples for IgG anti-*M. hyopneumoniae* were analyzed with Fisher’s test (*p* < 0.05). For other proportions that involved multiple comparisons (mortality, pneumonia index, and cough index), confidence intervals were obtained using the Wilson interval to establish differences based on the 95% confidence interval. All statistical analyses were performed using the software R Project for Statistical computing [41], the R Studio^®^ v. 3.5.1 (R Studio^®^, Boston, MA, USA) and the packages: agricolae [42], lm4 [43], love [44], emmeans [45], car [46], nortest [47], mass [48], and ggplot2 [49].

## 6. Conclusions

Our findings suggested that better results in relation to individual performance were achieved when using antibiotics pulses, and that protocols based only on valnemulin were more effective than the other molecules evaluated in the metaphylactic drug protocols involving macrolides administration. The protocols that used 5 and 10 mg/kg of valnemulin in sows and piglets proved to be the most promising in combining the maintenance and improvement of productivity, allowing a reduction in the total amount of antimicrobials administered. Furthermore, they may be related to an improvement in respiratory health.

## Figures and Tables

**Figure 1 antibiotics-11-00893-f001:**
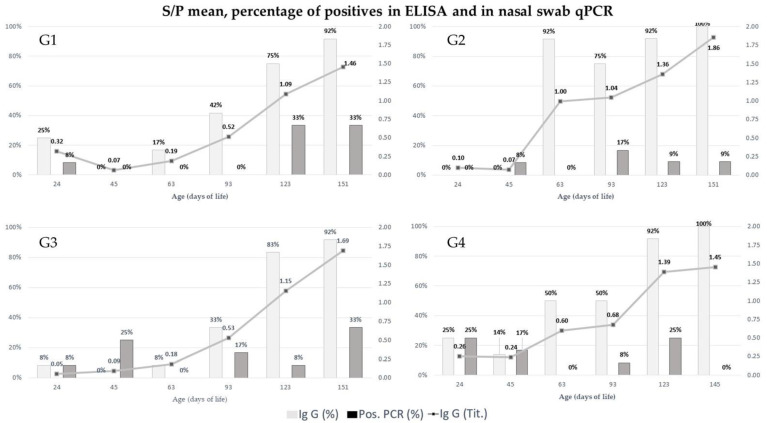
Mean value and percentage of serologically positive animals in the different groups (ELISA S/*P* value) for IgG anti-*M. hyopneumoniae* and prevalence of positive animals in nasal swab qPCR.

**Figure 2 antibiotics-11-00893-f002:**
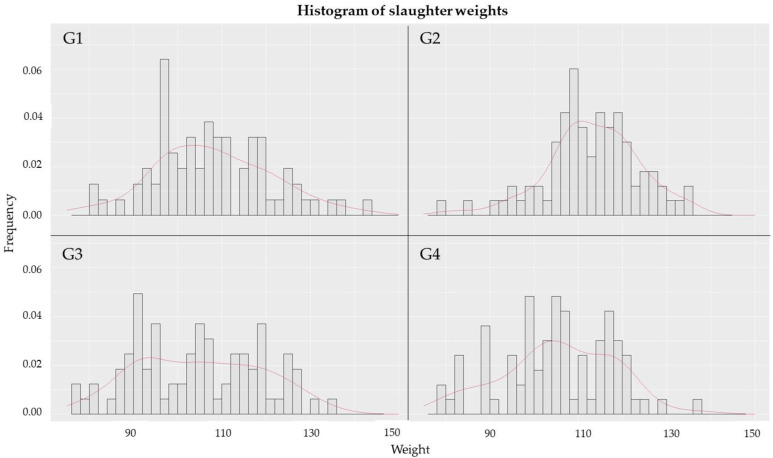
Weight histograms for G1, G2, G3 at 151 days and for G4 at 145 days of age.

**Figure 3 antibiotics-11-00893-f003:**
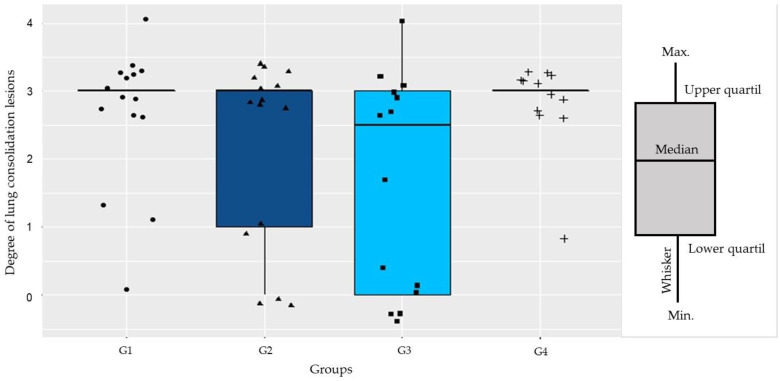
Degrees of injury among groups in the histopathological analysis. In the figure, individual values of each of the observations were represented as dots for G1, triangles for G2, squares for G3 and crosses for G4.

**Figure 4 antibiotics-11-00893-f004:**
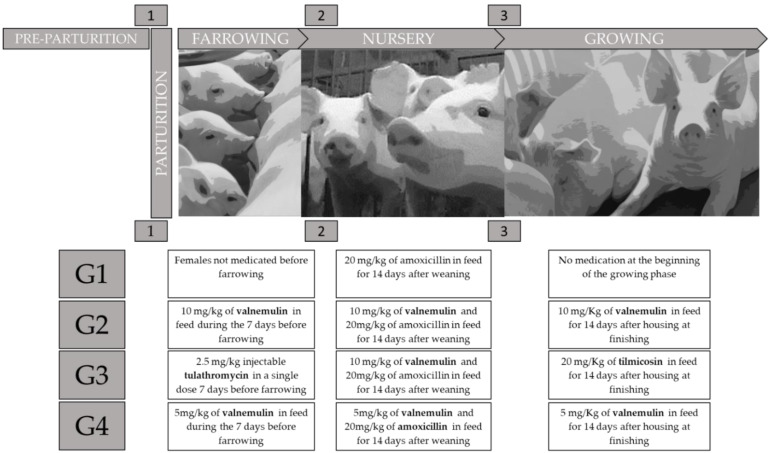
Schematics of medication protocols used in animals from G1 to G4.

**Table 1 antibiotics-11-00893-t001:** Mean S/*P* value for IgG anti-Mhyo between groups over time for each time point (T) from T1 to T6 (151 days of age for G1, G2 and G3 pigs and 145 days of age for G4 pigs).

Points	Group	Mean *	Median Standard Error	*p* Value
T1 (24 days)	G1	0.32 ^a^	0.14	2.35 × 10^−1^
	G2	0.10 ^a^	0.04	
	G3	0.05 ^a^	0.04	
	G4	0.26 ^a^	0.14	
T2 (45 days)	G1	0.07 ^a^	0.03	2.7 × 10^−1^
	G2	0.07 ^a^	0.02	
	G3	0.09 ^a^	0.03	
	G4	0.24 ^a^	0.13	
T3 (63 days)	G1	0.19 ^b^	0.05	2.3 × 10^−5^
	G2	0.10 ^a^	0.15	
	G3	0.18 ^b^	0.05	
	G4	0.60 ^ab^	0.18	
T4 (93 days)	G1	0.52 ^a^	0.17	1.5 × 10^−1^
	G2	1.04 ^a^	0.19	
	G3	0.53 ^a^	0.19	
	G4	0.68 ^a^	0.16	
T5 (123 days)	G1	1.09 ^a^	0.22	6.4 × 10^−1^
	G2	1.36 ^a^	0.22	
	G3	1.15 ^a^	0.19	
	G4	1.39 ^a^	0.15	
T6 (151 days)	G1	1.46 ^a^	0.21	2.9 × 10^−1^
	G2	1.86 ^a^	0.11	
	G3	1.69 ^a^	0.20	
	G4	1.45 ^a^	0.14	

* For each time point, means followed by the same letter are not significantly different by Tukey’s parametric test (*p* < 0.05).

**Table 2 antibiotics-11-00893-t002:** Weight at weaning (T1), at the end of nursery (T2), and at slaughter time (T3) for the four different groups.

Points	Group	Mean *	Standard Error	*p* Value
T1 (weight at 25 days)	G1	6.20 ^b^	0.17	1 × 10^−5^
	G2	7.05 ^a^	0.14	
	G3	6.01 ^b^	0.14	
	G4	6.47 ^b^	0.14	
T2 (weight at 63 days)	G1	20.64 ^b^	0.42	7 × 10^−6^
	G2	22.42 ^a^	0.40	
	G3	20.66 ^b^	0.48	
	G4	23.35 ^a^	0.43	
T3 (G1, G2 and G3 weight at 151 days, G4 weight at 145 days)	G1	107.18 ^b^	1.50	8 × 10^−5^
	G2	113.33 ^a^	1.28	
	G3	104.95 ^b^	1.47	
	G4	103.42 ^b^	1.86	
T3 (G4 estimated weight at 151 days)	G1	107.18 ^ab^	1.50	8 × 10^−3^
	G2	113.33 ^a^	1.28	
	G3	104.95 ^b^	1.47	
	G4	108.29 ^ab^	2.34	

* For each time point, means followed by the same letter are not significantly different by Tukey’s parametric test (*p* < 0.05).

**Table 3 antibiotics-11-00893-t003:** Average daily weight gain (ADWG) of the four groups in the nursery, finishing phase, and weaning to slaughter phase.

ADWG (kg)	Group	Mean *	Standard Error	*p* Value
Nursery	G1	0.39 ^b^	0.01	4 × 10^−6^
	G2	0.41 ^b^	0.01	
	G3	0.39 ^b^	0.01	
	G4	0.45 ^a^	0.01	
Finishing	G1	1.01 ^b^	0.01	3 × 10^−3^
	G2	1.07 ^a^	0.01	
	G3	0.99 ^b^	0.01	
	G4	1.01 ^b^	0.01	
Between 25d and 151d, considering the G4 up to 145d	G1	0.82 ^b^	0.01	
	G2	0.87 ^a^	0.01	1 × 10^−2^
	G3	0.81 ^b^	0.01	
	G4	0.84 ^ab^	0.01	

* For each phase, means followed by the same letter are not significantly different by Tukey’s parametric test (*p* < 0.05).

**Table 4 antibiotics-11-00893-t004:** Feed conversion of the four groups in the nursery, finishing, and weaning to slaughter phases.

Food Conversion (FC) (kg)	Group	Mean *	Standard Error	*p* Value
Nursery	G1	2.01 ^a^	0.05	5 × 10^−9^
	G2	1.76 ^b^	0.05	
	G3	2.11 ^a^	0.06	
	G4	1.69 ^b^	0.04	
Finishing	G1	2.12 ^a^	0.03	4 × 10^−2^
	G2	2.02 ^b^	0.03	
	G3	2.12 ^ab^	0.03	
	G4	2.03 ^ab^	0.03	
Between 25d and 151d, considering the G4 up to 145d	G1	2.12 ^a^	0.03	3 × 10^−4^
	G2	1.97 ^b^	0.02	
	G3	2.14 ^a^	0.03	
	G4	1.97 ^b^	0.03	

* For each phase, means followed by the same letter are not significantly different by Tukey’s parametric test (*p* < 0.05).

**Table 5 antibiotics-11-00893-t005:** Mean quantification of *M. hyopneumoniae* DNA (copies/µL) in lung tissue at slaughter for the four different groups.

Molecular Quantification of *M. hyopneumoniae* in Lung			
Group	Mean *	Standard Error	*p* Value
G1	1.34 × 10^5 a^	4.84 × 10^4^	4 × 10^−2^
G2	6.74 × 10^4 ab^	2.90 × 10^4^	
G3	3.15 × 10^4 b^	9.72 × 10^3^	
G4	2.41 × 10^4 b^	1.06 × 10^4^	

* Means followed by the same letter are not significantly different by Tukey’s parametric test (*p* < 0.05).

**Table 6 antibiotics-11-00893-t006:** Mean quantification of *M. hyopneumoniae* DNA (copies/µL) in bronchoalveolar lavage fluid at slaughter in the four different groups.

Molecular Quantification of *M. hyopneumoniae* in BALF			
Group	Mean *	Standard Error	*p* Value
G1	2.32 × 10^6 a^	9.33 × 10^5^	4 × 10^−1^
G2	5.67 × 10^6 a^	2.23 × 10^6^	
G3	3.57 × 10^6 a^	1.05 × 10^6^	
G4	3.33 × 10^6 a^	9.55 × 10^5^	

* Means followed by the same letter are not significantly different by Tukey’s parametric test (*p* < 0.05).

## Data Availability

Not applicable.

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
