# Peer review of "Chemotherapeutic Strategies with Valnemulin, Tilmicosin, and Tulathromycin to Control Mycoplasma hyopneumoniae Infection in Pigs"

_antibiotics, 2022, doi:10.3390/antibiotics11070893_

Round 1

Reviewer 1 Report

Chemotherapeutic strategies with Valnemulin, Tilmicosin, and 2 Tulathromycin to control Mycoplasma hyopneumoniae infection in pigs

No: antibiotics-1728070

Journal: Antibiotics

Article Type: Original Article

The current study addresses a very important issue and is of general interest to the readers of the journal: the rational use of antimicrobials to decrease AMR threat, new protocols that could control the Swine Enzootic Pneumonia (SEP), which is highly prevalent worldwide with financial losses for the swine industry.

The authors designed a randomized study with four groups (one control group) with 32 females in total to evaluate the effects of the application of different drug protocols in the control of the occurrence of Enzootic Pneumonia in commercial pig production and to provide data on a metaphylactic drug treatment strategy with the antimicrobials tilmicosin, valnemulin, and tulathromycin against M. hyopneumoniae infection in pigs.

I found the paper very interesting, well designed, executed and written. I am convinced that the authors had made a great effort in this study. I congratulate the authors and my suggestions are only in some minor aspects.

Suggestions:

Table 1: This table is different from others (column “difference” should be after standard error). I suggest not having a column to evidence the differences and write as “supersript”. Explain the number “1” after p-value. And this should be consistent with 3 decimal points in all (or scientific notation).

Figure 1: It is missing the designation of the groups.

Table 2: Since no p-value is significant and group differences, I suggest removing or send this table to supplementary files and explain trough the text the results.

Table 8: It is confusing. Since table 8 is related to the means of pulmonary consolidation areas and there is no p-value is significant and group differences, I suggest removing or send this table to supplementary files and explain trough the text the results. I did not find the results written in lines 231-236 in table 8.

Table 9: Since the p-value is not significant and none group differences, I suggest removing or send this table to supplementary files.

Figure 3: It needs improvement. The legend is missing (different marks, median, superior and inferior limits).

Suggestion: To compile all the results in one larger table. It helps us to verify all the results.

Also, I miss the figure S1 in the article. I think it is important to have the protocol used in the study to guide us.

Discussion: I miss having a first paragraph with the summary of the main results.

I think it is good that the authors are exploring new protocols with the rational use of antibiotics. However, my main concern relates to the fact that even with antibiotics (emergency use, last resource) it was not possible to completely control the disease. In the last paragraph, the authors talk about the importance of sanitary measures, which I believe are at the heart of the problem. A study that includes review and modification of management measures aligned with some ATB protocol would be very interesting. It is not about an issue of choosing only the right protocol. Changes in management may involve high initial costs for farms, but they bring long-term benefits: better outcomes, better animal welfare, and reduced use of antibiotics and thus fewer problems related to resistance and residues. This is a global threat and in some regions of the world, the metaphilatic use of antibiotics is being banned and replaced by good management procedures.

Author Response

Comment: Table 1: This table is different from others (column “difference” should be after standard error). I suggest not having a column to evidence the differences and write as “superscript”. Explain the number “1” after p-value. And this should be consistent with 3 decimal points in all (or scientific notation).

Answer: Thank you for the suggestions. Changes were made to the tables, removing the column “difference” and adding the letters together with the means/medians. In addition, we changed the formatting of p-values in the tables to scientific notation as suggested (pages 3, 5, 7, 8, 9, and supplementary material).

Comment: Figure 1: It is missing the designation of the groups

Answer: Right. We added the identification of the groups in the respective graphs (page 4).

Comment: Table 2: Since no p-value is significant and group differences, I suggest removing or sending this table to supplementary files and explain trough the text the results

Answer: We agree. The table has been transferred to the supplementary materials and the content was described in the text of the results section (lines 102 to 106, pages 3 and 4).

Comment: Table 8: It is confusing. Since table 8 is related to the means of pulmonary consolidation areas and there is no p-value is significant and group differences, I suggest removing or send this table to supplementary files and explain trough the text the results. I did not find the results written in lines 231-236 in table 8.

Answer: We agree, table 8 was transferred to the supplementary file and the main information was added to the text of the results section (lines 270 to 274, page 9).

Comment: Table 9: Since the p-value is not significant and none group differences, I suggest removing or sending this table to supplementary files.

Answer: We agreed with the suggestion and that table was transferred to the supplementary files.

Comment: Figure 3: It needs improvement. The legend is missing (different marks, median, superior and inferior limits).

Answer: Improvements were made in Figure 3 as suggested, adding a legend to aid in the interpretation of the results presented using the boxplot.

Comment: Suggestion: To compile all the results in one larger table. It helps us to verify all the results.

Answer: Thank you for the suggestion, it evaluated the possibility of inserting a table containing all the data, as it would really be interesting to observe the results together. However, after trying, we found that it would not be possible to build a table that was easily understood, since there are parametric data, represented by the mean, and non-parametric data, represented by the median. In addition to the fact that measurements such as S/P IgG and quantification in the nasal swab are repeated measures, the quantification of Mhyo in the lung, degree of lung injury, and quantification of M. hyopneumoniae in the BALF, in turn, are measures recorded in a single time point. The productive measures (ADWG and weight) differ in terms of interpretation, although they are related to the same period, one being recorded over the period (ADWG and FC) and the other a point measure (weight). In addition, the measurement points for production data do not correspond exactly to the measurement points for laboratory analysis. Consequently, the tables have been kept separate, and we believe it is easier to interpret that way.

Comment: Also, I miss the figure S1 in the article. I think it is important to have the protocol used in the study to guide us.

Answer: We agreed with the suggestion and figure S1 was transferred to the manuscript (Figure 4, page 15).

Comment: Discussion: I miss having a first paragraph with the summary of the main results.

Answer: We agree with the suggestion and added a paragraph with a summary of the main results obtained in the present study.

Comment: I think it is good that the authors are exploring new protocols with the rational use of antibiotics. However, my main concern relates to the fact that even with antibiotics (emergency use, last resource) it was not possible to completely control the disease. In the lat paragraph, the authors talk about the importance of sanitary measures, which I believe are at the heart of the problem. A study that includes review and modification of management measures aligned with some ATB protocol would be very interesting. It is not about an issue of choosing only the right protocol. Changes in management may involve high initial costs for farms, but they bring long-term benefits: better outcomes, better animal welfare, and reduced use of antibiotics and thus fewer problems related to resistance and residues. This is a global threat and in some regions of the world, the metaphilatic use of antibiotics is being banned and replaced by good management procedures.

Answer: Thank you for the observation. The strategies for controlling and eradicating M. hyopneumoniae in production are complex and involve different aspects, but they undoubtedly bring countless benefits to swine production. Hopefully, future studies will collaborate to elucidate the effectiveness of M. hyopneumoniae eradication protocols in commercial production, and it will provide essential information to the scientific community and commercial production. In the present study, considering that it was a propriety where M. hyopneumoniae infection occured endemically, and the groups evaluated were a small part of the animals of the property during the period in which the experiment was conducted, it would be challenging to control the infection or drastically reduce the number of infected animals, by only changing antimicrobial molecules. One of the points to be considered concerning this aspect is that only based on the detection of M. hyopneumoniae in the swab, lung, and lavage, it is not possible to conclude whether one or another drug protocol was effective in controlling the infection, for the simple fact that the detection of M. hyopneumoniae by qPCR does not distinguish live bacteria from dead bacteria, being a technique based on the presence or absence of the agent's genetic material. We believe that in this sense it is important to consider all variables analyzed together, and with that, it is evident that the results obtained in the present study are promising as they indicate an increase in productivity in herds, associated with the rational use of antimicrobial molecules, which in itself it is positive and promotes technical/scientific basis for future studies with even more innovative objectives.

Reviewer 2 Report

Dear Author, 

the manuscript needs to be improved. An extensive English revision is required. Materials and methods should be detailed (i.e. cough examination).

Author Response

Comment: the manuscript needs to be improved. An extensive English revision is required. Materials and methods should be detailed (ie cough examination).

Answer: Thank you for the comments, and we agree. An extensive revision of the English was carried out, and changes were made to the material and methods section to detail the methodology (page 16, lines 580 to 585; 606 to 608; page 17, lines 224 to 231; page 18, lines 676 to 682).

Reviewer 3 Report

In this study, Giovani M Stingelin et al. evaluate the use of antimicrobial drugs to minimize the impact of Mycoplasma hyopneumoniae infection in pigs, using 32 pregnant female pigs and their litters involving three experimental groups with different treatments and a control group.

The authors mentioned different treatments and they conclude that the promising treatment is the one with Valnemulin. However, in the experimental design at some point during the evaluation, all the animals received a dose of Valnemulin, only group 3 (G3) received Tulathromycin at the farrowing phase and Tilmicosin at growing phase. Having this in mind, it is risky to make the conclusion that Valnemulin treatment is better and promising than the others, since there is no clear difference between treatments. In my opinion, this is the biggest problem in this report. The authors should be clearer about why they decide to use this strategy when designing the animal experiment, and mention this in the report.

By the other hand, the results should be shown in a table where all the data from all the animals is compared. For an instant, at the IgG evaluation is very easy to mask the differences between animals. I do understand that you are trying to compare between groups the presence of IgG, however it is important to show the antibody titer of each pig, not only the mean of each group. Again, these results can be masking the real result from each animal/treatment. The same problem with the rest of the results.

It will be very interesting to see results using the three different protocols with the differences between drugs. Since maybe the pharmacokinetics between the drugs should be a problem, it will be necessary to improve the experimental design, maybe using another antimicrobial drug.

Author Response

Referee No. 3

Comment: The authors mentioned different treatments and they conclude that the promising treatment is the one with Valnemulin . However, in the experimental design at some point during the evaluation, all the animals received a dose of Valnemulin , only group 3 (G3) received Tulathromycin at the farrowing phase and Tilmicosin at the growing phase. Having this in mind, it is risky to make the conclusion that Valnemulin treatment is better and promising than the others, since there is no clear difference between treatments. In my opinion, this is the biggest problem in this report. The authors should be clearer about why they decide to use this strategy when designing the animal experiment, and mention this in the report.

Answer: In fact, all groups, except group 1, received valnemulin at some point, and perhaps in the way we have described, we have given the impression that valnemulin was the determining factor for the differences observed for group 2 compared to the others, therefore changes have been made to the text of the manuscript (lines 465 to 475). Based on the results, we believe that the protocol applied to group 2 was more efficient than the other protocols, although the dose and the molecule used are not the only ones responsible for this.

It must be considered that the treatment of the sows may have been a determining factor in the dynamics of infection since no animal in G2 was positive for M. hyopneumoniae in the nasal swabs until weaning. In breeding females, valnemulin was applied only to groups 2 and 4 (at different doses), and possibly it was more effective than tulathromycin, applied to females in group 3, taking into account that they are molecules with distinct pharmacokinetic characteristics, with the maximum pulmonary concentration of valnemulin (4h) occurring in a shorter time when compared to tulathromycin (24h) (Stingelin et al., 2022; Yuan et al., 2011; Benchaoui et al., 2004)

Another crucial point in the authors' understanding is the differences between treatments in the growing phase, as pleuromutilins are described as being up to 30 times more efficient than macrolides in controlling M. hyopneumoniae, although both have good oral absorption (>90%) (Giguère et al., 2013) and similar mechanisms of action (Spinosa et al. 2011)

Although, it is not possible to affirm that the differences observed in the performance indicators are associated with treatment with valnemulin only considering the type of molecule used. It is evident that different protocols, although involving the same molecule, presented different results, mainly in performance indicators. In that regard, different results observed, possibly due to the substitution of valnemulin for tulathromycin or tilmicosin ( both considered less effective than pleuromutilin) and the similarity between the results comparing groups 2 and 4 (different doses of valnemulin) are relevant information from the point of view of the authors of the present study.

Comment:  By the other hand, the results should be shown in a table where all the data from all the animals is compared. For an instant, at the IgG evaluation is very easy to mask the differences between animals. I do understand that you are trying to compare the presence of IgG between groups, however it is important to show the antibody titer of each pig, not only the mean of each group. Again, these results can be masking the real result from each animal/treatment. The same problem with the rest of the results.

Answer: We agree that measures of central tendency in some cases may not be representative of the set of observations, depending on several factors such as the nature of the data (ordinal, continuous, categorical), order of magnitude, and even variability. Therefore, we seek to pay attention to statistical analysis when working with the correct identification and distinction of parametric and non-parametric data. Evaluation of normality and homoscedasticity were performed for all data in the present study to define which of the measures of central tendency would best represent each data set. Also, means and medians were expressed together with the standard error (for mean and median) that aid the reader to understand the variability of the data. Furthermore, in the case of categorical data, as in the classification of lung injury in degrees, we sought to illustrate the dispersion of the data using the boxplot which, from our point of view, provides other measures that can lead the reader in understanding the results presented. As suggested a table containing the individual S/P values has been added to the supplementary material.

Comment: It will be very interesting to see results using the three different protocols with the differences between drugs. Since maybe the pharmacokinetics between the drugs should be a problem, it will be necessary to improve the experimental design, maybe using another antimicrobial drug.

Answer:  Currently, there is a limited number of molecules that do not belong to the Group III of antimicrobials classified by the WHO (Quinolones, Cephalosporins, Polymyxins, and Phosphonic Acids), which is a category of drugs, and although being used in animals, is considered a last resort due to the possibility of dissemination of resistance genes that cause impacts on human health. And even less, of drugs that are efficient against M. hyopneumoniae. Within this aspect, the molecules used in the present study are reported as promising molecules due to the results observed in vitro (Távio et al. 2014) and for their characteristics of being of high oral absorption when administered in tfeed (macrolides: tilmicosin (90 %), pleuromutilins: valnemulin (>95%)) and superior in this aspect when compared to other molecules: tylosin (22.5%), neomycin and colistin (0%), doxycycline (45%), lincomycin (51%) (Giguere et al., 2013).

Although there are few data in the literature that attest to the positive effect of the in vivo use of pleuromutilins and macrolides in the treatment of respiratory diseases in swine, those antimicrobial classes are widely used to control Swine Enzootic Pneumonia (SEP) in herds. The choice of molecules was because they are currently used for this purpose and are practically the only resources for a metaphylactic treatment of M. hyopneumoniae infection. Also, in the literature, it is reported that there are differences between the classes of antimicrobials, despite having mechanisms of similar action. Therefore, we believe that the use of other molecules from other antimicrobials classes would even be interesting from the point of view of results, although it may not be representative of the current commercial swine production scenario. Therefore, in this sense, we believe that the protocols used are aligned with the reality in the field. Although there are similarities between the protocols concerning the use of valnemulin, the data demonstrate a difference between the protocols of G2 and G4 compared to G3 and G1, which may indicate that not only the choice of the molecule is important, as well the strategy of metaphylactic treatment as a whole.

BENCHAOUI, HA et al. Pharmacokinetics and lung tissue concentrations of tulathromycin in swine. Journal of veterinary pharmacology and therapeutics, vol. 27, no. 4, p. 203-210, 2004.

YUAN, LG et al. A physiologically based pharmacokinetic model for valnemulin in rats and extrapolation to pigs. Journal of veterinary pharmacology and therapeutics, vol. 34, no. 3, p. 224-231, 2011.

STINGELIN, GM et al. Pharmacological aspects of valnemulin, tilmicosin, and tulathromycin applied to the treatment of respiratory diseases in swine. 2022 (work prepared for publication but not submitted)

Round 2

Reviewer 3 Report

The authors had been answered my questions. Still I have doubts about the experimental design, but the knowledge contribution is important.

Author Response

Comment: The authors had been answered my questions. Still I have doubts about the experimental design, but the knowledge contribution is important.

Thank you for the comment. Based on what was discussed in the reports (first and second round) and in the editor's comments, a paragraph describing the study limitations has been added to the discussion section (page 12).
